# Extraction and Characterization of Lysozyme from Salted Duck Egg White

**DOI:** 10.3390/foods11223567

**Published:** 2022-11-09

**Authors:** Xinjun Yao, Tianyin Du, Jun Guo, Weiqiao Lv, Benu Adhikari, Jicheng Xu

**Affiliations:** 1College of Biological and Food Engineering, Anhui Polytechnic University, Wuhu 241000, China; 2College of Biology and Food Science, Suzhou University, Suzhou 234000, China; 3College of Engineering, China Agricultural University, Beijing 100083, China; 4School of Science, RMIT University, Melbourne, VIC 3083, Australia

**Keywords:** salted duck egg white, lysozyme, purification, enzymatic properties

## Abstract

Salted duck egg white (SDEW), as the main by-product in the production process of salted egg yolk, has not been effectively used as a food resource because of its high salt concentration. This study creatively used isoelectric point precipitation, ultrafiltration, and cation exchange to separate and purify lysozyme from SDEW and preliminarily explored the enzymatic properties of lysozyme. The results showed that the relative molecular weight of lysozyme was about 14 KDa, and the specific activity of lysozyme reached 18,300 U/mg. Lysozyme had good stability in the temperature range of 30 °C to 60 °C and pH of 4 to 7. Metal ions, Fe^2+^, Cu^2+^, and Zn^2+^, strongly inhibited lysozyme activity. Different surfactants showed certain inhibition effects on lysozyme from SDEW, among which glycerin had the strongest inhibitory effect. This study aimed to provide a theoretical reference for industrial purification and production of lysozyme from SDEW.

## 1. Introduction

Salted duck egg white (SDEW) is an important part of salted duck eggs. As a by-product of salted duck egg yolk production, several tens of thousands of tons of SDEW are discarded annually [1,2]. The amino acid composition of SDEW is a high-price, high-quality protein that can be fully utilized by the human body [3,4]. On the one hand, the abandoned SDEW causes the waste of high-quality protein resources; on the other hand, its spoilage and decomposition significantly pollute the surrounding environment and water sources [5,6]. Therefore, a good research topic is the study of the comprehensive utilization of SDEW. Lysozyme (LZ) is one of the main bioactive components in SDEW that can catalyze the hydrolysis of β-1, 4-glycosidic bond between N-acetylmuramic acid and N-acetylglucosamine in the bacterial cell wall, and has no adverse effect on human cells without cell walls [7]. It is a natural antibacterial substance that widely exists in nature and is the most abundant in the egg whites of birds and poultry. It accounts for about 3.50% of the total egg white protein [8]. Lysozyme has been widely used in food, medicine, the chemical industry, and other fields [9]. At present, most of the commercial lysozyme on the market is obtained through its separation and purification from egg white. However, there are few literature reports on the separation and purification of lysozyme from SDEW.

The main extraction methods of lysozyme include the direct crystallization method [10], ion exchange method [11], ultrafiltration method [12], two-phase system separation method [13], reverse micelle extraction method [14], and affinity membrane chromatography [15]. Arica and Bayramoğlu [16] used reactive blue 4 and reactive red 120 dye ligand immobilized complex membranes to purify lysozyme from egg white, with a recovery rate of 16% and 72%, respectively. The activated red 120 immobilized membranes showed a high adsorption capacity and selectivity for purifying lysozyme from egg white. However, it is difficult to achieve industrial mass production using this technology, owing to its excessive experimental operation and relatively high cost. Wolman et al. [17] used non-covalently bound chitin retained between silicon oxide matrix layers to prepare a composite biosorbent for the purification of lysozyme from undiluted egg white. The results showed that 64% of lysozyme was removed from the egg white. This process is characterized by its low cost and high absorption rates, but the disadvantages of cumbersome preparation of the composite biosorbent and time-consuming production processes are a limitation in practical production. Dembczyński and Białas [18] obtained lysozyme with a total recovery rate of 47.5%, purification factor of 10.5, and specific activity of 34,188 U/mg by combining the two-phase distribution system with membrane separation technology. In this process, lysozyme was extracted using binary separation technology, and the purity of lysozyme was high. However, it is still necessary to further explore the recovery of residual egg white protein and realize the expanded extraction process of lysozyme.

The aim of this study was to creatively use isoelectric point precipitation, ultrafiltration, and cation exchange to separate and purify lysozyme from SDEW. In addition, we preliminarily explored the enzymatic properties of this lysozyme to provide an innovative idea for industrial purification and production of lysozyme from SDEW.

## 2. Materials and Methods

### 2.1. Raw Materials

SDEWs were provided from Anqing Tianhe Food Co., Ltd. (Anqing, China). Lysozyme standard and test kits were purchased from Nanjing Jiancheng Biology Co., Ltd. (Nanjing, China). Low molecular weight protein markers were procured from Shanghai Ruichu Biotechnology Co., Ltd. (Shanghai, China). D152 macroporous weak acid cation exchange resin was obtained from Zhengzhou Ainuo Technology Co., Ltd. (Zhengzhou, China). A sodium dodecyl sulfate–polyacrylamide gel electrophoresis (SDS-PAGE) gel preparation kit was purchased from Wuhan Xavier Biotechnology Co., Ltd. (Wuhan, China). Ammonium sulphate, hydrochloric acid, and phosphate buffer were purchased from Sinopharm Chemical Reagent Co., Ltd. (Shanghai, China).

### 2.2. Extraction Process of Lysozyme

#### 2.2.1. Pretreatment of SDEW

For SDEW sample pretreatment, SDEW was filtered using 2–4 layers of gauze. The filtered SDEW was stirred with a magnetic stirrer at 106× *g* for 20 min. For the pretreatment of ion exchange resin [19], D152 microporous weak acid cation exchange resin was repeatedly cleaned with distilled water to remove impurities. The resin was soaked in 1000 mL hydrochloric acid (1 mol/L) for 12 h. Next, the resin was rinsed with distilled water until the pH of the rinsed distilled water was 9. The above soaking and rinsing steps were then repeated once more.

#### 2.2.2. Protein Removal from Salted Duck Egg White

This study used isoelectric point precipitation to remove proteins from SDEW. The egg white was mixed with phosphate buffer solution (0.20 mol/L, pH = 6.8) at a volume ratio of 1:4. The mixture was placed in a refrigerator at 4 °C for 24 h. Next, the supernatant was collected using a centrifuge at 4 °C and 8603× *g* for 30 min.

#### 2.2.3. Isolation of Lysozyme from SDEW by Ultrafiltration

The molecular weight of the hybrid proteins in SDEW was mostly greater than 30 kDa. Polyethersulphone (PES) membrane with a molecular weight of 30 kDa was used to remove most of the hybrid proteins, and lysozyme in SDEW was fully retained by the ultrafiltration membrane. The experiment was carried out at room temperature (25 °C), the operating pressure was 0.35 MPa, and the phosphate buffer solution (0.20 mol/L) with a pH of 8.5 was diluted 3 times continuously [20]. The permeate was collected for further purification.

#### 2.2.4. Desalting and Concentration of Lysozyme Filtrate Using Ultrafiltration

The lysozyme filtrate was desalted and concentrated by ultrafiltration using a PES membrane with a molecular weight of 5 kDa. The operating pressure of the experiment was 0.35 Mpa, and the desalting and concentration processes were carried out using ultrafiltration at room temperature (25 °C). The permeate was collected for further purification.

#### 2.2.5. Purification of Lysozyme by Cation Exchange Adsorption

Purification of lysozyme was performed using the cation exchange adsorption method according to the procedure described by Show et al. [21], with slight modifications. The desalted lysozyme filtrate was stirred and mixed with D152 ion exchange resin to cover the resin completely. After stirring for 6 h, the mixture was left to stand and then the supernatant of the mixture was discarded. Ammonium sulfate solution (10%) was added to the resin, stirred for another 1 h, and the eluent was collected. The above elution steps were repeated thrice to collect the filtered eluent. The eluent was centrifuged in a frozen centrifuge at 5000× *g* for 20 min. Lysozyme was obtained by freezing and drying the supernatant to form a dry powder.

#### 2.2.6. Optimization of Cation Exchange Adsorption of Lysozyme

Based on the single factor test, four factors, including resin dosage (A), adsorption time (B), eluent concentration (C), and elution time (D), were selected, and the Box–Behnken test with four factors and three levels was adopted (Table 1). The optimum extraction process was determined by the lysozyme content.

### 2.3. Determination of Lysozyme Concentration

The lysozyme content was determined according to the method of Amaly et al. [22], with slight changes. The experiment used 0.90% NaCl solution as the reference solution. An analytical balance was used to weigh 0.50 g of standard lysozyme accurately and this was added to 0.90% NaCl to prepare a 0.50 mg/mL solution. Next, 1, 2, 3, 4, and 5 mL of lysozyme solution were placed in 10 mL stopped colorimetric tubes. The NaCl solution (0.90%) was added to each of these tubes until the final volume was 10 mL. The absorbance of the sample solution was measured successively at 281 nm. Three groups of parallel experiments were conducted to take the average value and draw the standard curve. The standard curve equation was y = 2.4257x + 0.0094 (R^2^ = 0.9973, Figure 1). The error of the slope was 0.0822 and the error of interception was 0.0136. The lysozyme concentration was calculated using the regression equation of the standard curve.

The formula for calculating the lysozyme content in SDEW was as follows:(1)$$e=\frac{c\times a\times V}{w}$$
where *c* is the concentration of lysozyme in the eluent (mg/mL), *a* is the dilution ratio, *V* is the volume of eluent (mL), and *w* is the volume of SDEW used for extraction (mL).

The lysozyme yield in SDEW was calculated using the following formula:(2)$$y=\frac{m}{M}$$
where *y* is the lysozyme yield (%), *m* is the lysozyme quantity (mg), and *M* is the mass of SDEW used for extraction (mg).

### 2.4. Determination of Specific Activity of Lysozyme

The specific activity of lysozyme was determined by referring to the experimental method of Geng et al. [23], with slight changes. The enzyme solution (0.40 mL) and wall lytic micrococcus bacterial solution (2 mL) were added to the tube to be measured, and the time was recorded. After the reaction at 450 nm for 15 s and 75 s, the absorbances A_15_ and A_75_ were recorded. The specific activity of the enzyme was defined as one activity unit (U) that decreased by 0.001 compared with the OD_450_ value per min.
(3)$$E_{A}=\frac{1000\times (A_{15}-A_{75})}{E_{W}}$$
where E*_A_* is the specific activity of lysozyme (U/mg) and E*_w_* is the lysozyme content in the measured enzyme solution (mg).

### 2.5. Purity Identification of Lysozyme Samples

The purity of lysozyme from SDEW was determined using SDS-PAGE [24]. Lysozyme was dissolved in water at 100 °C for 10 min in a water bath. Next, 10 µL of standard protein and sample solutions was placed in the corresponding Eppendorf Tubes, and buffer solution (10 µL) was added to them, respectively. A 20 µL sample was added to each well, and electrophoresis was performed. The concentrations of separating and stacking gels were 12% and 5%, respectively, and the voltage was 100 V. After electrophoresis, the gel was fixed with a fixative containing acetic acid solution. The gel was then stained with Coomassie bright blue R250 and decolorized with decolorization solution (glacial acetic acid:methanol:water = 2:1:17). The electrophoresis results were recorded using a gel imager. The dried electrophoretic film was then analyzed using an optical density scanning software. The purity of lysozyme was calculated according to the dye concentration.

### 2.6. Microstructure Analysis Using Scanning Electron Microscopy (SEM)

The microstructure of lysozyme from SDEW was analyzed according to the method described by Huang et al. [25] with some modifications. The freeze-dried lysozyme was installed on the SEM screw roots using double-sided conductive tape. The lysozyme samples were coated with a thin layer of gold in a sputtering device, and the apparent morphology was examined using an SEM.

### 2.7. Analysis of Secondary Structure Characteristics

Fourier transform infrared spectrometer (FTIR, IS10, Thermo Nicolet Corp., Madison, WI, USA) was used to analyze the secondary structure of lysozyme [26]. The freeze-dried lysozyme samples (1 mg) of SDEW were mixed with 100 mg of dried potassium bromide. The mixture was then ground into a uniform powder and pressed into thin slices. The thin slices were examined using an infrared spectrometer (4000–400 cm^−1^). The signals were collected from 32 scans at a resolution of 4 cm^−1^. OMNICV 8.0 software was used to analyze the spectral data. The secondary structure of lysozyme was quantitatively analyzed by peak fitting of the FTIR spectrum. The relative percentages of different secondary structures were calculated according to the peak area.

### 2.8. Determination of Enzymatic Properties of Lysozyme from SDEW

#### 2.8.1. Thermal Stability and Acid–Base Stability

The lysozyme obtained from SDEW was dissolved in phosphate buffers with different pH values (3, 4, 5, 6, 7, 8, and 9) to attain 20 μg/mL solutions. All the solutions were maintained in a water bath at 25 °C for 30 min to study their optimum pH. Then, the relative enzyme activity of the solution was measured every 30 min. A total of six measurements were performed to compare the changes in enzyme activity at different pH values and different times to evaluate the pH stability of enzyme activity.

Next, the lysozyme was then dissolved in phosphate buffer at optimum pH to produce a 20 μg/mL solution. All the solutions were held for 30 min at different temperatures (20, 30, 40, 50, 60, 70, 80, and 90 °C) to determine the optimal reaction temperature. Then, the relative enzyme activities of all the solutions were measured at different temperatures at 30 min intervals. In both experiments, the highest enzyme activity was set at 100%, and the other enzyme activities under different pH values and temperatures were denoted as the highest relative ratio of enzyme activity. A total of six measurements were performed to compare the changes in enzyme activity at different temperatures and times and to evaluate the temperature stability of the enzyme activity.

#### 2.8.2. Determination of the Effect of Metal Ions on Lysozyme Activity

The solution of lysozyme from SDEW was prepared with phosphate buffer at a 20 μg/mL mass concentration. Next, different metal ions (Na^+^, K^+^, Ca^2+^, Fe^2+^, Cu^2+^, Mn^2+^, Zn^2+^, and Mg^2+^) were added to the enzyme solution, and the content of each metal ion in the solution was 0.01 mol/L. At the same time, a group of enzyme solutions without any metal ions was used as the blank control group, and the enzyme activity was set at 100%. After all the enzyme solutions were kept in a water bath at 25 °C for 30 min, the relative enzyme activity of each experimental group was determined.

#### 2.8.3. Determination of Effect of Surfactants on Lysozyme Activity

The SDEW lysozyme solution was prepared with phosphate buffer at a 20 μg/mL mass concentration. Next, glycerol, Span20, Span40, Span80, Tween20, Tween40, and Tween80 were added to the enzyme solution at a concentration of 1.0 mg/L for each surfactant. At the same time, a group of enzyme solutions without surfactants was used as the blank control group, and the enzyme activity was set at 100%. After all the enzyme solutions were maintained in a water bath at 25 °C for 30 min, the relative enzyme activity of each experimental group was determined.

### 2.9. Statistical Analysis

The SPSS 20.0 software (IBM, Chicago, IL, USA) and Design Expert 10.0.7 were used for the analysis of variance of the samples in the study. Significant differences were determined by Duncan’s multiple comparison test (*p* < 0.05).

## 3. Results and Discussion

### 3.1. Optimal Conditions for Lysozyme Extraction from SDEW Using Single-Factor Test

It can be observed from Figure 2a that the lysozyme concentration did not increase when the resin dosage exceeded 40 mL, indicating that there was no residual lysozyme in the ultrafiltrate. Therefore, the optimal resin dosage was 40% of the volume of the ultrafiltrate. As shown in Figure 2b, the lysozyme concentration gradually increased with the extension of adsorption time. When the adsorption time reached 6 h, the concentration of lysozyme remained stationary. Therefore, the optimal adsorption time was 6 h. As shown in Figure 2c, its elution capacity gradually improved with the increase in ammonium sulfate ((NH_4_)_2_SO_4_) concentration. When the (NH_4_)_2_SO_4_ concentration exceeded 12%, the lysozyme concentration increased slightly. The possible reason was that partial salting out of the protein occurred at high salt concentrations, resulting in increased turbidity of the solution and absorbance. Therefore, the optimal concentration of (NH_4_)_2_SO_4_ was 10%. As shown in Figure 2d, the lysozyme concentration gradually increased with the extension of elution time. After the elution time reached 60 min, the lysozyme concentration remained unchanged; therefore, the elution time was set at 60 min.

### 3.2. Optimization of Experimental Conditions Using the Box–Behnken Test

The Box–Behnken test was further conducted to optimize the experimental conditions. The experimental design and results are shown in Table 2 and Table 3. The multiple regression equation obtained by fitting was as follows: enzyme concentration = 2.57 + 0.22A + 0.18B + 0.097C + 0.067D + 0.081 AB − 0.038AC + 0.058AD + (5.263E − 0.003) BC + 0.16BD + 0.11CD − 0.73A^2^ − 0.66 B^2^ − 0.55 C^2^ − 0.50 D^2^. (A: resin dosage, B: adsorption time, C: (NH_4_)_2_SO_4_ concentration; D: elution time).

Design Expert 10.0.7 software was used to analyze the data, and the experimental results are shown in Table 3. The P value of the model was less than 0.001, reaching a significant level, indicating that the model fits the actual situation well. However, lack-of-fit tests were not significant. All the factors, A, B, A^2^, B^2^, C^2^, and D^2^, reached significant levels. The correlation coefficient R^2^ of the equation was 0.9312, indicating that the model could explain 93.12% of the data, and the equation model had a high degree of fit.

To obtain the highest lysozyme concentration, an appropriate cross-selection of adsorption and elution conditions was required. Response surface diagrams can directly reflect the effects of various factors and interactions on enzyme activity. Therefore, three interaction terms with high significance were selected to make a response surface map (Figure 3). According to the response surface diagram and variance analysis, the order of influence on lysozyme activity was BD > DD > AB.

The above multiple regression equations were derivatized. When the response value (lysozyme concentration) was at its maximum, the resin dosage (A) was 35.81 mL, the adsorption time (B) was 6.32 h, the concentration of (NH_4_)_2_SO_4_ (C) was 1.02 mol/L, and the elution time (D) was 62.26 min per 100 mL egg white filtrate. In such a condition, the theoretical prediction of lysozyme concentration was 2.61 mg/mL. The optimal conditions were verified by three repeated tests, and the lysozyme content was 2.59 mg/mL, with an error of ±1% compared to the theoretical value. The experimental results showed that the established process parameters were reliable and feasible.

According to the optimum process conditions obtained from the response surface optimization experiment, the eluent was ultrafiltered, dialyzed, and freeze-dried in an ultrafiltration centrifuge tube. The yield of the lysozyme was 0.36%, and the enzyme activity was 18,300 U/mg.

### 3.3. SDS-PAGE Profile of Lysozyme from SDEW

The final lysozyme powder was subjected to SDS-PAGE. The SDS-PAGE profile of lysozyme in SDEW is shown in Figure 4. The bands in the standard lane showed molecular weight distribution, with relatively concentrated and clear bands, mainly the 14 kDa (lysozyme) band. It can be observed that the extracted samples were mainly lysozyme. The results were consistent with the SDS-PAGE profiles reported by Santos et al. [27]. The results showed that the lysozyme extracted from SDEW had reached electrophoretic purity and could be used for further experiments.

### 3.4. Microstructure of Lysozyme from SDEW

The microstructures of lysozyme in hen egg white and *SDEW* were observed using scanning electron microscopy (Figure 5). According to the observation at 10k× magnification using SEM, the surface of lysozyme (A) in the control group was smooth, the arrangement was relatively loose, and the shape mainly was oval or round rod. The surface of lysozyme (B) in the experimental group was rough and compact, and its shape was an irregular round or round rod. The reason for this result may be attributed to the fact that under the influence of a high salt concentration in salted duck eggs, the surface structure of lysozyme changes to a certain extent, resulting in a rough surface. There were some residual protein particles in the experiment, and their gel properties made the lysozyme distinct and more compact. In general, the microstructure of lysozyme extracted from SDEW in this study was relatively clear and distinct, and the shape of the lysozyme was similar to that of natural lysozyme. The purity of lysozyme was relatively high, which could be used in subsequent experimental studies.

### 3.5. FTIR Spectrum of Lysozyme from SDEW

The FTIR spectrum of lysozyme from SDEW is shown in Figure 6. The spectrum shows good characteristic absorption peaks, in particular 1652 cm^−1^ attributed to the amide I band (1600–1700 cm^−1^) and 1234 cm^−1^ to the amide III band (1220–1330 cm^−1^). Two characteristic absorption peaks contain secondary structure information for lysozyme, including β-fold, α-helix, irregular curl, and β-turn [28]. The amide I band (1600–1700 cm^−1^) was related to the C=O stretching vibration in the FTIR spectrum. By second derivative analysis and peak fitting, one α-helix and three β-turn absorption peaks can be obtained. By the same method, four characteristic peaks can be obtained for the amide III bands (Table 4). The relative content of the SDWE lysozyme secondary structure was analyzed using a combination of amide I and amide III bands. The result shows that the relative content of SDEW lysozyme was 0.06 for β-sheet, 0.10 for α-helix, 0.82 for β-turn, and 0.02 for irregular curl, respectively. The results were similar to the secondary structure of natural lysozyme and suggested that the hydrogen bonding network of SDEW lysozyme was not disrupted after desalination. Therefore, lysozyme extracted from SDEW has as good an application as any other natural lysozyme source.

### 3.6. Enzymatic Properties of Lysozyme from SDEW

#### 3.6.1. Effect of pH and pH Stability on Lysozyme Activity

As shown in Figure 7, the effect of pH on the activity of lysozyme extracted from SDEW increased initially and then decreased. When the pH was 7, the lysozyme activity reached its maximum. Therefore, the optimal pH of lysozyme in SDEW was about 7. This change in lysozyme activity may be because SDEW lysozyme was positively charged under alkaline conditions at pH < 10. As the pH of the solution approached the isoelectric point of lysozyme (pI > 10), the charge of lysozyme reduced and had less opportunity to bind to the substrate, resulting in reduced activity. Thus, changes in pH resulted in changes in the charged state of lysozyme, which in turn resulted in changes in enzyme activity [29].

As shown in Figure 8, the relative activity of lysozyme remained above 85% in the range of pH 4 to 7, showing good stability. When pH > 7, lysozyme activity gradually decreased with the increase in treatment time. In conclusion, lysozyme from SDEW was stable under acidic and neutral conditions. However, the activity of lysozyme decreased significantly under the condition of high pH. The possible reason was that the main chain conformation of lysozyme in an acidic and neutral environment was only minimally affected by the change in acidity, and its structure was relatively stable. However, extreme changes in pH could cause the dissociation of internal molecular groups, resulting in the denaturation of lysozyme [30].

#### 3.6.2. Effect of Temperature and Temperature Stability on Lysozyme Activity

As observed in Figure 9, with an increase in reaction temperature, the activity of lysozyme from SDEW showed an increasing trend at first, followed by a decrease. The relative activity of lysozyme increased slowly with the increase in temperature from 20 °C to 60 °C. It may be because the high temperature causes the intermolecular movement to accelerate, promoting increased enzyme activity. However, when the temperature exceeded 60 °C, the lysozyme was gradually inactivated. Therefore, the optimal reaction temperature of lysozyme was 60 °C.

As shown in Figure 10, the lysozyme activity of SDEW was relatively stable at a lower temperature (20 °C to 60 °C). The activity of lysozyme decreased with the prolongation of treatment time. The decrease in lysozyme activity was accelerated with the increase in temperature. The reason for this phenomenon might be that with the increase in temperature, the thermal denaturation rate of lysozyme was accelerated, leading to the rapid inactivation of lysozyme.

#### 3.6.3. Effect of Metal Ions on Lysozyme Activity

Metal ions and surfactants play an important role in the process of lysozyme performing various functions. Understanding the effect of metal ions on enzyme activity is helpful for enzyme application and modification. It can be observed from Figure 11 that Na^+^ and Mg^2+^ activated lysozyme. However, Fe^2+^, Cu^2+^, and Zn^2+^ strongly inhibited lysozyme activity. Other metal ions (Mn^2+^, K^+^, and Ca^2+^) had no significant effect on lysozyme activity. In conclusion, most metal ions had little effect on lysozyme activity, except for a few metal ions with a strong inhibitory effect.

#### 3.6.4. Effect of Surfactants on Lysozyme Activity

As shown in Figure 12, different surfactants had slight inhibitory effects on the lysozyme from SDEW. Among them, Span20, Span40, Span80, Tween20, Tween40, and Tween80 had little effect on the enzyme activity, and the relative enzyme activity remained above 80%. The activity of lysozyme treated with glycerol decreased to 75%. The surfactants combine with lysozyme to form a complex. The hydrophilic group of the surfactant binds to the enzyme through hydrogen bonding [31], reducing the lysozyme activity.

## 4. Conclusions

In this study, isoelectric point precipitation, ultrafiltration, and cation exchange were innovatively used to isolate and extract lysozyme from SDEW, and its enzymatic properties were explored. The results showed that lysozyme with high purity was obtained, wherein the yield was 0.36%, and the enzyme activity was 18,300 U/mg. The optimal pH of lysozyme from SDEW was 7, and the stability was good in the pH range of 4 to 7. The optimal reaction temperature of lysozyme was 60 °C, and its thermal stability was good at low temperatures. However, the lysozyme activity decreased significantly after 60 °C. Moreover, the metal ions, Na^+^ and Mg^2+^, activated lysozyme activity, while Fe^2+^, Cu^2+^, and Zn^2+^ inhibited the lysozyme activity. Other metal ions (Mn^2+^, K^+^, and Ca^2+^) had no significant effect on SDEW lysozyme activity. Furthermore, different surfactants exhibited certain inhibitory effects on lysozyme from SDEW, and glycerol showed the strongest inhibitory effect. This work could help to comprehensively utilize SDEW resources, find a high extraction rate and purity of lysozyme, and provide an innovative way to achieve high efficiency and yields of lysozyme after its extraction and separation from SDEW.

## Figures and Tables

**Figure 1 foods-11-03567-f001:**
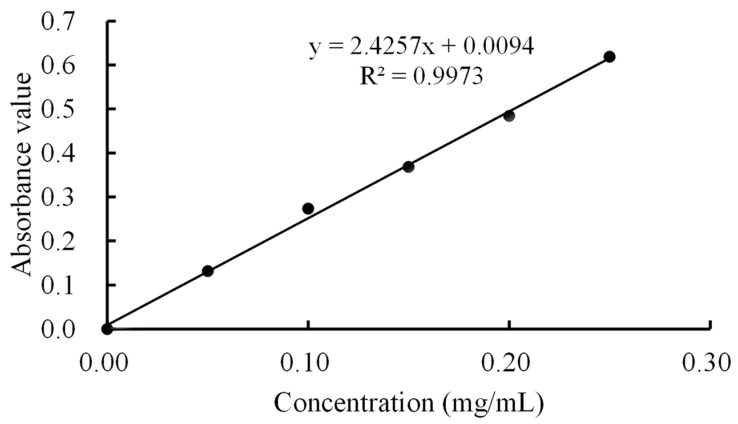
Standard curve of lysozyme solution.

**Figure 2 foods-11-03567-f002:**
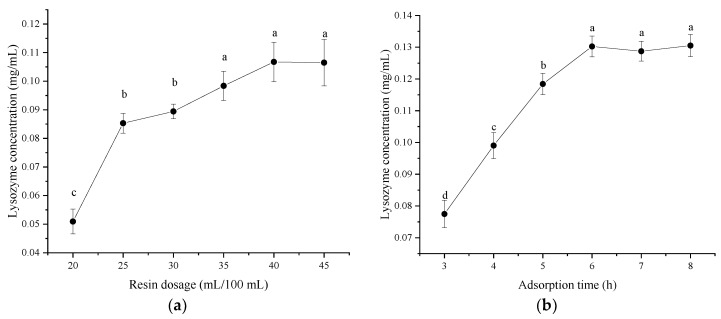
Effect of single factor on the concentration of lysozyme from salted duck egg white (SDEW). (**a**) Resin dosage, (**b**) adsorption time, (**c**) ammonium sulfate concentration; (**d**) elution time. Different lowercase letters (**a**–**d**) in the same graph represent significant differences (*p* < 0.05).

**Figure 3 foods-11-03567-f003:**
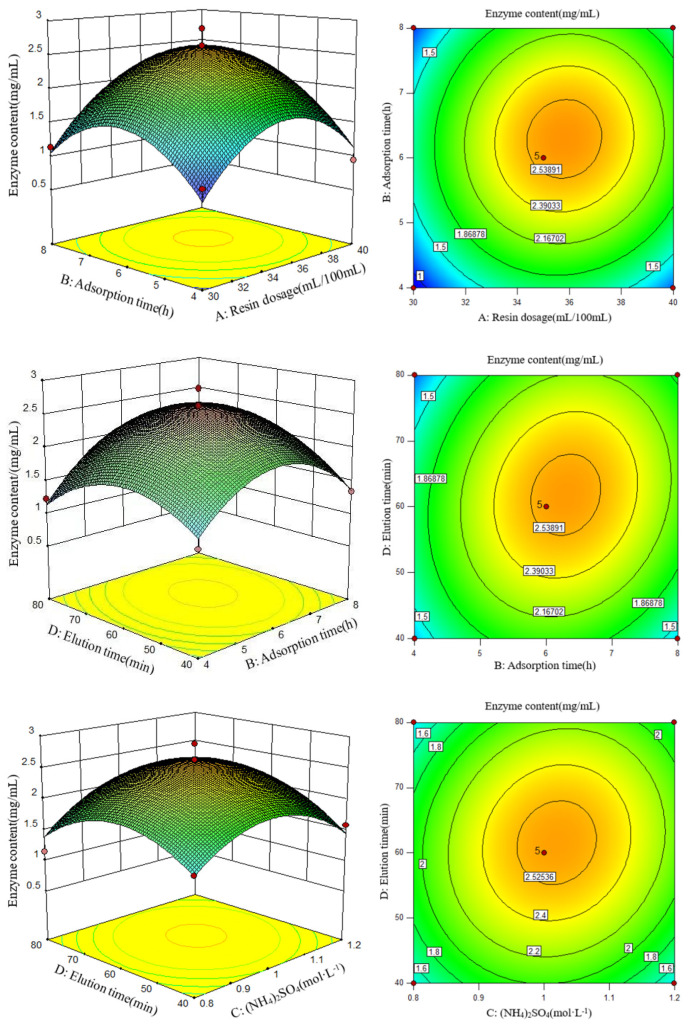
Response surface and contour plots for the effects of (**A**) resin dosage, (**B**) adsorption time, (**C**) ammonium sulfate concentration, and (**D**) elution time on the content of lysozyme obtained from SDEW.

**Figure 4 foods-11-03567-f004:**
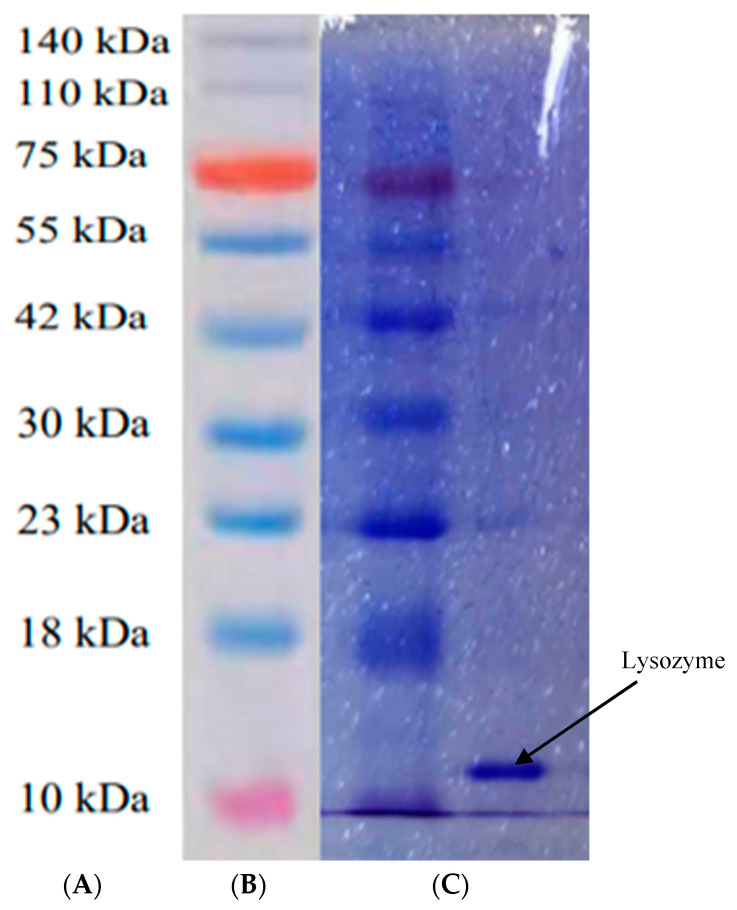
SDS-PAGE profile of lysozyme from SDEW. (**A**) Protein molecular quality standard, (**B**) standard protein Marker; (**C**) sample.

**Figure 5 foods-11-03567-f005:**
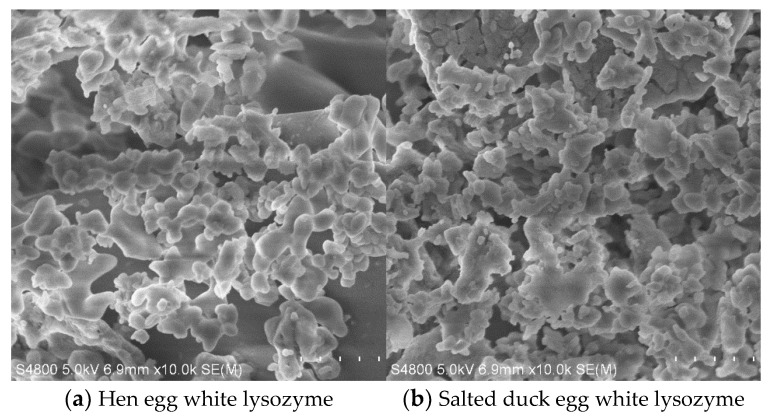
Comparison of lysozyme from hen egg white and SDEW.

**Figure 6 foods-11-03567-f006:**
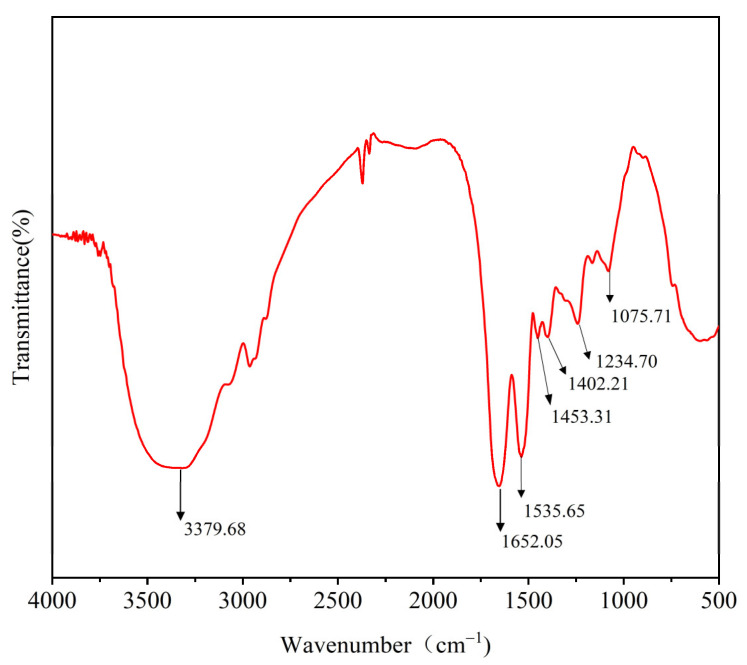
Infrared spectrum of lysozyme from SDEW.

**Figure 7 foods-11-03567-f007:**
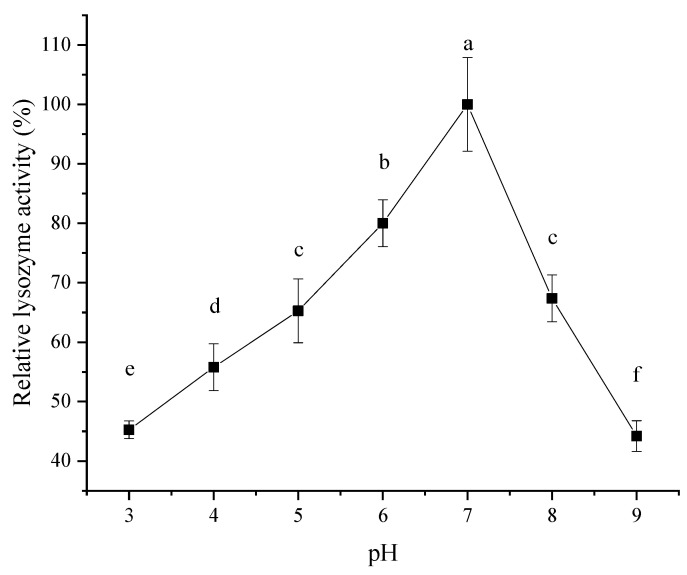
Effect of pH on activity of lysozyme from SDEW.

**Figure 8 foods-11-03567-f008:**
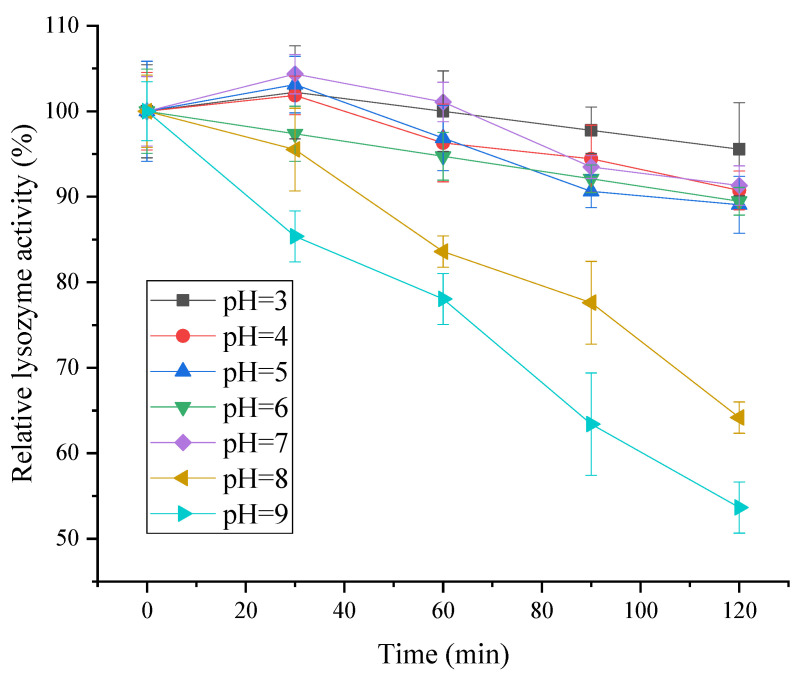
pH stability of lysozyme from SDEW.

**Figure 9 foods-11-03567-f009:**
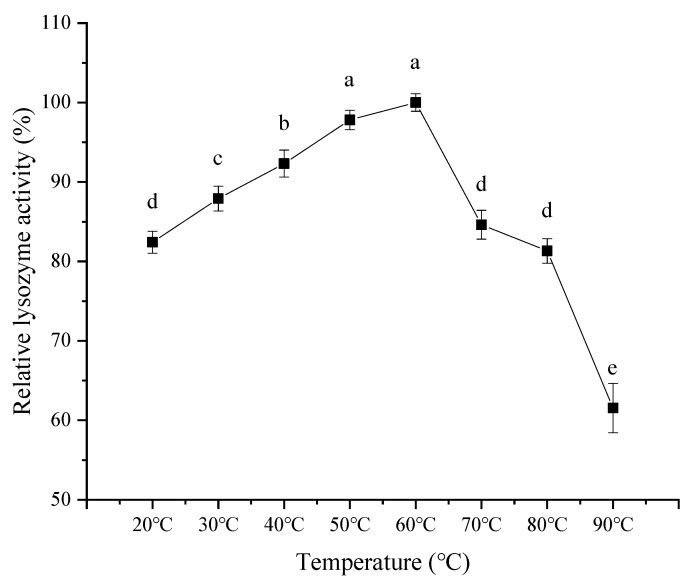
Effect of temperature on activity of lysozyme from SDEW.

**Figure 10 foods-11-03567-f010:**
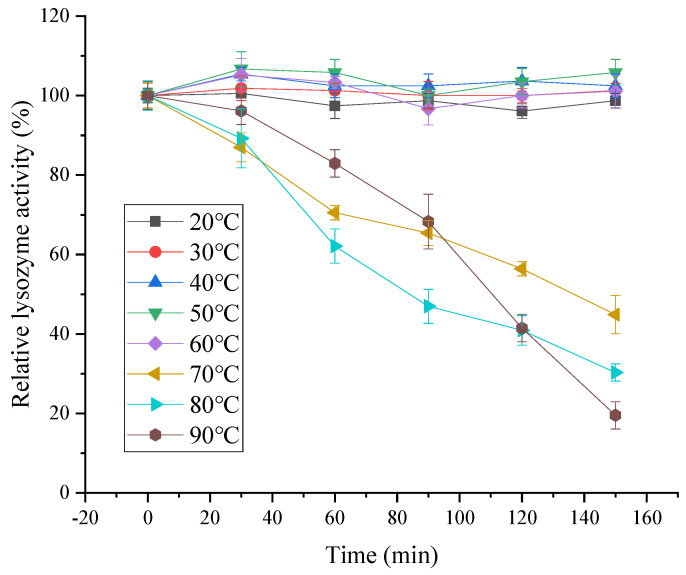
Temperature stability of lysozyme from SDEW.

**Figure 11 foods-11-03567-f011:**
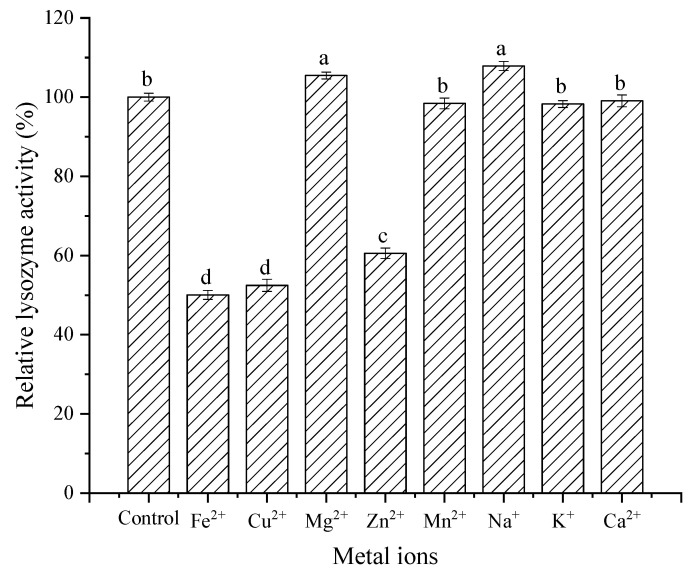
Effect of metal ions on activity of lysozyme from SDEW.

**Figure 12 foods-11-03567-f012:**
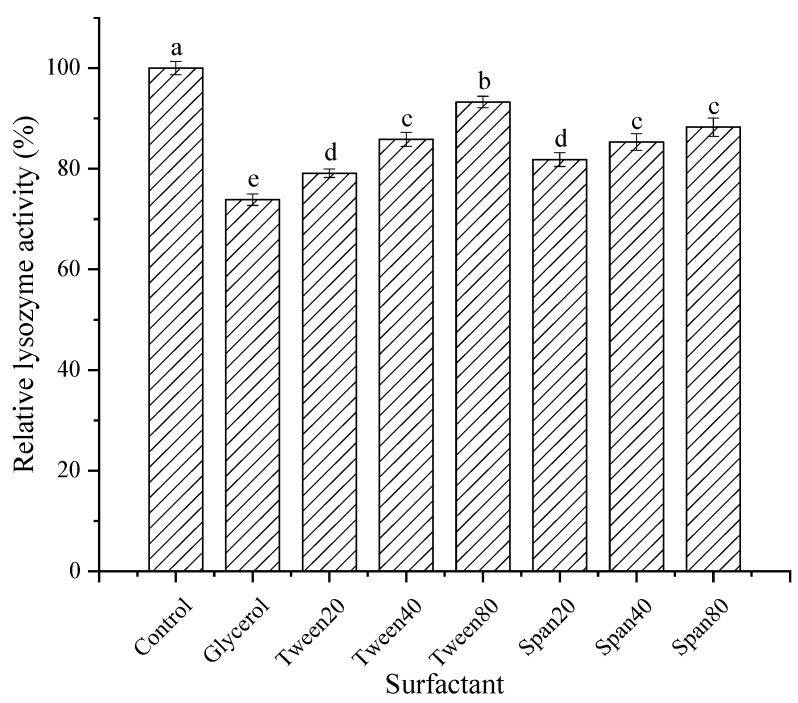
Effect of surfactants on activity of lysozyme from SDEW.

**Table 1 foods-11-03567-t001:** Factors and their levels employed in the Box–Behnken experimental design.

Level	Resin Dosage (A) (mL/100 mL)	Adsorption Time (B) (h)	Eluent Concentration (C) (mol/L)	Elution Time (D) (min)
−1	30	4	0.80	40
0	35	6	1.00	60
1	40	8	1.20	80

**Table 2 foods-11-03567-t002:** Actual values of the factors obtained using Box–Behnken experiment.

Number	Resin Dosage A (mL/100mL)	Adsorption Time B (h)	Ammonium Sulfate Concentration C (mol/L)	Elution Time D (min)	Enzyme Concentration (mg/mL)
1	40	8	1.00	60	1.38
2	35	6	1.00	60	2.36
3	35	4	1.00	80	1.23
4	35	6	0.80	80	1.15
5	35	4	1.20	60	1.32
6	35	4	0.80	60	1.10
7	40	6	1.20	60	1.63
8	35	6	1.00	60	2.45
9	35	8	1.00	40	1.35
10	30	6	0.80	60	0.95
11	40	6	1.00	40	1.52
12	35	6	1.20	40	1.59
13	35	6	1.00	60	2.88
14	35	6	1.20	80	1.68
15	30	8	1.00	60	1.13
16	35	6	1.00	60	2.63
17	30	6	1.20	60	1.06
18	40	4	1.00	60	0.94
19	35	8	0.80	60	1.41
20	35	8	1.00	80	2.04
21	40	6	0.80	60	1.67
22	35	6	0.80	40	1.47
23	35	4	1.00	40	1.17
24	30	4	1.00	60	1.02
25	35	6	1.00	60	2.52
26	40	6	1.00	80	1.79
27	30	6	1.00	40	1.04
28	30	6	1.00	80	1.07
29	35	8	1.20	60	1.65

**Table 3 foods-11-03567-t003:** Analysis of variance of the results obtained using Box–Behnken experiment.

Source	Square Sum	Degrees of Freedom	Mean Square	*f*-Value	*p*-Value	Significance
Model	7.80	14	0.56	13.53	<0.0001	***
A	0.60	1	0.60	14.48	0.0019	***
B	0.40	1	0.40	9.70	0.0076	***
C	0.11	1	0.11	2.76	0.1188	NS
D	0.054	1	0.054	1.31	0.2712	NS
AB	0.027	1	0.027	0.64	0.4354	NS
AC	0.0058	1	0.005807	0.14	0.7129	NS
AD	0.013	1	0.013	0.32	0.5791	NS
BC	0.00011	1	0.0001108	0.002691	0.9594	NS
BD	0.100	1	0.100	2.43	0.1416	NS
CD	0.045	1	0.045	1.08	0.3155	NS
A^2^	3.48	1	3.48	84.63	<0.0001	***
B^2^	2.87	1	2.87	69.66	<0.0001	***
C^2^	1.94	1	1.94	47.23	<0.0001	***
D^2^	1.59	1	1.59	38.62	<0.0001	***
Error	0.58	14	0.041			
Lack of fit	0.42	10	0.042	1.04	0.5287	NS
Net error	0.16	4	0.040			
Total error	8.37	28				

Note: NS indicates no significant difference; *** represents a significant difference (*p* < 0.01).

**Table 4 foods-11-03567-t004:** Secondary structure analysis of salted duck egg white lysozyme, including relative content of the main form of the secondary structure of salted duck egg white lysozyme.

**Amide I Amide**				**Amide III Amide**		
	**Peak Position** **/cm^−1^**	**Peak Area**	**Assignment**	**Peak Position** **/cm^−1^**	**Peak Area**	**Assignment**
SDEW lysozyme	1656.16	24.11	α-helix	1221.08	17.59	β-sheet
	1675.92	55.49	β-turn	1268.46	5.83	Irregular curl
	1688.77	94.61	β-turn	1275.59	2.15	β-turn
	1699.26	83.86	β-turn	1295.28	5.60	α-helix
**Relative Amount**						
	**Amide I Amide**		**Amide III Amide**	**Amide I Amide + Amide III Amide**		
β-sheet	—		0.56	0.06		
α-helix	0.09		0.19	0.10		
β-turn	0.91		0.07	0.82		
Irregular curl	—		0.18	0.02		

## Data Availability

The data generated during the present study are available from the corresponding author upon reasonable request.

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
