# Peer review of "Extraction and Characterization of Lysozyme from Salted Duck Egg White"

_foods, 2022, doi:10.3390/foods11223567_

Round 1

Reviewer 1 Report

Lysozyme has been widely used in food, medicine, the daily chemical industry, and other fields so  development industrial purification and production of lysozyme from SDEW are usefull

The article developes the thechnology  of obtaining lysozyme from duck white eggs. It is of undoubted interest

According to the reviewer methods of characterization would be more convincing if they were supplemented by SAXS and molecular dynamics methods  

Reviewer 2 Report

In the manuscript, Yao and coworkers introduced a method for lysozyme separation from salted duck egg white, including isoelectric point precipitation, ultrafiltration, and cation exchange. The authors presented an overall study on the separation procedure and on lysozyme catalytic performance (against temperature, pH, and in the presence of ions and surfactants). The paper provides good results. However, there are a few things to be addressed, please see below:

1. The language needs some improvement.

2. Paragraph 3.5, Page 7:

a) The authors write: “In the amide I band, the absorption peaks were seen at 1652 cm-1, 1535 cm-1, and 1453 cm-1, consistent with the absorption peaks of the lysozyme standard.” Amide I band corresponds to the region of 1600-1700 cm-1 (as also mentioned by the authors). Thus, only the peak at 1652 cm-1 is correlated to the Amide I band. The peak at 1535 cm-1 could correspond to the Amide II region (1500-1600 cm-1). Moreover, there is no peak at 1453 cm-1 as seen from Figure 6. Please make the appropriate band assignments.

b) the authors also write: “The hydrogen bond network of lysozyme was not damaged after desalting. Thus, the secondary structure of lysozyme extracted from SDEW was the same as that of natural lysozyme from other sources”. This declaration is misleading. The FTIR spectrum itself is not enough to evaluate changes in the secondary structure of the enzyme. To be sure that the secondary structure of the protein was not damaged, a more detailed analysis of the Amide I band should be included, by providing the % contribution of each secondary element (through peak deconvolution and analysis). In the Materials and Methods section (Paragraph 2.7) the authors claimed that they indeed made this kind of analysis. However, the results are missing. Please provide the analysis.

3. Paragraph 3.6.1, Page 7: the authors write “This change in lysozyme activity may be because the nature and quantity of lysozyme charge changes with pH value”. Lysozymes have a pI>10, thus till pH=9.0, lysozyme should be positively charged. Could the authors further explain, in a simple way, their declaration, to be more obvious to the reader?

4. Paragraph 3.6.3-Title: Pleas correct PH to pH

Reviewer 3 Report

The submitted manuscript reports an extraction method for the lysozyme from salted duck egg white and its characterization.

Below, I am listing some points that justify my opinion.

1)    English should be improved.

2)    Introduction: Authors refer “…there are certain limitations in its production”. Please clarify what kind of limitations are involved.

3)    Raw materials: “All the other reagents were analytically pure”. Please complete this sentences, including the reagents involved and the producer companies.

4)    Pretreatment of SDEW:

a) “…for 20 min at a speed that did not cause foaming of egg white.” Please, indicate the speed value or the speed interval values.

b)    “…the resin was rinsed with distilled water to a pH of about 9.” Please explain how the pH is measured in the resin.

5)    Optimization of cation exchange adsorption of lysozyme:

a)    “The lysozyme content in the eluent was determined by resin dosage, adsorption time, eluent concentration, and eluent time.” Please clarify this sentence. Probably “was determined by” should be replaced by “is dependent from”.

b)    “The lysozyme content was used as an index to evaluate the single-factor test results.” This sentence should also be rewritten in order to be clarified.

c)    “Based on the single factor test, four factors, including resin dosage (A), adsorption time (B), eluent concentration (C), and elution time (D), were selected, and the Box-Behnken test with four factors and three levels was adopted (Table 1).” This sentence is basically the repetition of the first sentence of this paragraph.

6)    Determination of lysozyme concentration:

a)    “Next, 1, 2, 3, 4, and 5 mL solutions were placed in a 10 mL colorimetric tube with a stopper, respectively.” The word “respectively” does not make sense in this sentence. Please rewrite the sentence to be clear.

b)    The calibration curve equation should be introduced in the text. It should present the errors associated with the slop and the interception for a specific confidence level.

c)    “…m is the lysozyme quality…” Please revise this expression. It seems to be “…m is the lysozyme quantity…”

7)    Purity identification of lysozyme samples: “… glacial acetic acid-methanol”. Please refer the mixture ratio.

8)    Thermal stability and acid-based stability: This section seems to be similar to section “Optimum temperature and pH value”. Please check it and make the necessary changes.

9)    Optimization of experimental conditions using the Box-Behnken test:

a)    “According to the single factor….of 60 min.” This information was previously referred.

b)    Please introduce the meaning of the letters (eg. A, B, C,….)

10) Effect of temperature on lysozyme activity: “ … temperatures from 60ºC to 80ºC.” This interval of temperature seems not to be in accordance with figure 8.

11) pH stability of lysozyme: “…remained above 80%.” Please check this value according with the results showed in figure 9.

12) Please explain why the effect of metals and the surfactants on the lysozyme was tested. Why the authors decided to test them instead of other compounds with known inhibitory effect on enzymes?

Round 2

Reviewer 3 Report

Below, I am listing some points that, in my opinion, still need to be changed and improved.

-  6 a) The sentence “Next, five 10 ml stoppered colorimetric tubes were taken and 1, 2, 3, 4 and 5 ml of lysozyme solution were placed inside each.” continues not clear. I suggest “Next, 1, 2, 3, 4, and 5 mL of lysozyme solution were placed in 10 mL stopped colorimetric tubes”.

-  6 b) the errors associated with the slop and the interception in the equation of the calibration curve, should be introduced. I suggest a confidence level of 95%.

8) The sections “Optimum temperature and pH value” and “Thermal stability and acid-base stability” continue to be similar. I suggest to join these two sections, since the thermal stability and acid-base stability are the reasons for the choice of the optimum temperature and pH.

-  9a) The sentence “According to the single factor test results” is not clear. Please revise the beginning of this paragraph.

12) No explanation about the reason for the study of the effect of metals and the surfactants on the lysozyme activity was introduced in the manuscript, as it was suggested before.
